# Fabrication of mm-Scale Complementary Split Ring Resonators, for Potential Application as Water Pollution Sensors

**DOI:** 10.3390/ma16155290

**Published:** 2023-07-27

**Authors:** Zacharias Viskadourakis, George Fanourakis, Evangelos Tamiolakis, Anna Theodosi, Klytaimnistra Katsara, Nikolaos Rafael Vrithias, Odysseas Tsilipakos, George Kenanakis

**Affiliations:** 1Institute of Electronic Structure and Laser (IESL), Foundation for Research and Technology—Hellas (FORTH), N. Plastira 100, Vasilika Vouton, GR-70013 Heraklion, Greece; georgefanourakis2@gmail.com (G.F.); annatheodosi@iesl.forth.gr (A.T.); clytokatsara@gmail.com (K.K.); n.vrithias@iesl.forth.gr (N.R.V.); 2Department of Materials Science Technology, University of Crete, GR-70013 Heraklion, Greece; vagostam@materials.uoc.gr; 3Department of Agriculture, Hellenic Mediterranean University, Estavromenos, GR-71410 Heraklion, Greece; 4Theoretical and Physical Chemistry Institute, National Hellenic Research Foundation, GR-11635 Athens, Greece; otsilipakos@eie.gr

**Keywords:** metamaterials, complementary metasurfaces, split ring resonators (SRRs), engraving method, water pollution sensors

## Abstract

Rectangular, millimeter-scale complementary split ring resonators were fabricated, employing the so-called Computer Numerical Control method, combined with a home-built mechanical engraver. Their electromagnetic performance was thoroughly investigated with respect to their dimensions in the frequency regime between 2 and 9 GHz via combining experiments and corresponding theoretical simulations, wherein a considerably effective consistency was obtained. Moreover, their sensing response was extensively investigated against various aqueous solutions enriched with typical fertilizers used in agriculture, as well as detergents commonly used in every-day life. Corresponding experimental results evidently establish the capability of the studied metasurfaces as potential sensors against water pollution.

## 1. Introduction

Water pollution can be defined as the process, during which harmful substances contaminate lakes, rivers, streams, aquifers, oceans etc., degrading the quality of water and rendering it toxic to humans as well as to the environment [1]. Water contamination could originate from several sources and human activities. Among others, two of the most common water contamination sources are the extensive use of fertilizers in agriculture and the wide and daily use of detergents [2,3].

Fertilizers, in general, contain nitrogen among other constituents, which is vital for plants since it is included in chlorophyll, the most important substance for photosynthesis [4,5]. Plants absorb the needed amount of nitrogen from the ground. However, heavy cultivation stemming from the dramatically increased human need for food reduces the nitrogen concentration of the ground. Consequently, such a lack of nitrogen is replaced via the use of chemical fertilizers. The wide use of fertilizers, however, turns out to be a severe water contamination source. Significant amounts of fertilizer residue remain in the ground. The rain leaches them and carries them into rivers, lakes, streams and aquifers, leading to direct contamination. Consequently, water from such sources cannot be used by humans, since it is harmful. Even more to the point, a large amount of these diluted residues are transferred to the seas, resulting in excessive plant and algal growth and consequent oxygen depletion. Such oxygen reduction leads to detriment of the local fish population, and finally, local ecosystems are severely disturbed or even destroyed. Such a process is widely known as eutrophication [6].

On the other hand, the extreme use of detergents is another source of water pollution. Most of the detergents used in everyday life contain chemicals and substances which can cause various pollution problems in water. Moreover, most of those chemicals cannot be decomposed or eliminated via public or private wastewater treatment, and therefore they are transferred to clean bodies of water (lakes, streams, rivers, etc.), resulting in water pollution. Therefore, contamination of the sea water via detergents contributes to the destruction of sea ecosystems [7,8,9].

Considering the above, the early and precise monitoring of pollution in water sources is of great importance for water quality improvement since global water supplies are becoming less while water demands are increasing dramatically, leading to a global water crisis [10]. In this context, sensors exhibiting high accuracy, sensitivity, and resolution are required [11,12]. Moreover, such sensors should be portable and responsive, so as to be placed into lakes, rivers, water reservoirs, and other bodies of water for quick and direct pollution measurements. Furthermore, the remote control/monitoring of those sensors would be beneficial for continuous and real-time observations of water resource contamination. So far, several interesting reports have been conducted in water pollution sensing based on various measurements, such as pH, conductivity, oxygen concentration, turbidity, spectroscopy, etc. [12,13,14,15,16].

Among these, microwave sensors have become an active research approach for water quality control with increasing interest. Microwave sensing technology possesses significant advantages since it is a non-destructive, low-cost, and low-power technique; it provides a direct and immediate response, and it can continuously provide real-time data in a remote manner [15]. In this context, metamaterial sensors functioning in the microwave regime have become a rather promising approach to water quality testing. In general, microwave metamaterials (and their two-dimensional counterparts, the so-called metasurfaces) are human-developed materials exhibiting extraordinary electromagnetic properties which cannot be observed in natural materials, such as perfect absorption, negative refractive index, visibility cloaking etc. [17,18]. The basic component of a metasurface, called a “meta-atom”, possesses specific shape and dimensions. The electromagnetic (EM) properties of such metasurfaces (MSs) are directly affected by those two factors. Additionally, the spatial distribution of the meta-atoms also affects the EM response of the MSs. Therefore, MSs can be used for several applications, such as antennas, splitters, isolators, modulators, electromagnetic shielding, and energy harvesters, as well as sensors [11,19,20,21,22,23].

A widely known and thoroughly studied metasurface is the so-called split ring resonator (SRR) [24,25], consisting of a metallic loop with a gap. Depending on their shape and size, SRRs exhibit resonance effects at certain frequencies, which are indicated by a sharp dip in their transmission spectrum. By inserting a tiny amount of a dielectric material into the gap, the geometric capacitance of the SRR is strongly affected, the resonance effect is disturbed, and such a disturbance is mainly exhibited via a shift in the resonance frequency. Since the dielectric material has a higher dielectric constant than air ($\varepsilon_{air}^{\prime }=1$), the shift is expected to be toward lower frequencies $\left(f_{res}\propto 1/2\pi \sqrt{LC}\right)$, enabling the sensing capability of the MS. In this context, there have been several recent investigations in which SRRs were used as water quality sensors, distinguishing between different kinds of organic and inorganic materials in aqueous solutions. Although, the main disadvantage of such SRR-like sensors is that the liquids used for sensing should be quite viscous in order to remain in the gap [26,27,28,29,30]. Interestingly, there are reports among these in which complementary SRRs (CSRRs; e.g., Figure 1a,b) are utilized as sensors [18,23,26,31,32,33]. Here, it should be noted that CSRRs behave similarly to the corresponding SRRs, although they are essentially the exact inverse counterpart of an SRR (an SRR consists of a metallic loop surrounded by a dielectric material, while the CSRR consists of a dielectric loop surrounded by metal).

Fabrication of conventional mm-scale SRRs includes a great variety of techniques, such as printed circuit board (PCB) [34], additive manufacturing [35], inkjet printing [36] etc. However, for the fabrication of high-performance CSRRs, the PCB method has mostly been used so far [28,29,30,31,37]. The PCB procedure is known as the most appropriate method for developing electronic circuits on substrates (such as FR4) and includes various development steps, namely chemical etching, milling, annealing, etc. Moreover, the corresponding infrastructure, needed, consists of several expensive instruments, while the personnel required must be highly trained. All of this increases the production cost when expanding from the laboratory to mass production scale. Hence, despite their high quality, the fabrication of PCB-printed CSRRs is limited by their increased production cost. To this point, alternative methods have been proposed, such as the computer numerical control (CNC) engraving technique. In general, CNC is used in the manufacture of metal, plastic, and wooden parts, while mills, lathes, routers, drills, grinders, etc. are some of the cutting tools that can be automated via CNC. Thus, CNC is dedicated to large-scale production since it reduces development time and man-hours, which consequently reduces production cost. In addition, engraving has been successfully used to develop mm-scale metasurfaces [38,39,40]. Therefore, the utilization of CNC engraving to develop CSRRs seems like quite an interesting approach. 

Considering all the above, in this study, the CNC method was employed in combination with an engraver in order to successfully fabricate rectangular CSRRs on metallized surfaces. Several rectangular CSRRs were printed, with various dimensions, changing their length, line, and gap width. The developed CSSRs were characterized in terms of their EM response; all of them exhibited characteristic resonance at certain frequencies, depending on their dimensions, indicating their effective EM performance. Moreover, their sensing capability was investigated for several water pollutants, including fertilizers as well as detergents. It was found that the sensing efficiency of the CSSRs was significant, even in cases of low-concentration solutions, indicating their high sensitivity. Finally, the sensing efficiency of the engraved CSRRs was studied against the degradation of fertilizers and detergents using a heterogeneous photocatalytic approach [41,42,43]. We hereby present evidence that the fabricated CSSRs could be effectively used as quite sensitive and high-resolution sensors of common water pollutants.

## 2. Materials and Methods

### 2.1. Metasurface Fabrication

The so-called rectangular complementary split ring resonator—CSRR—design was chosen for the purpose of the current study. In general, there is plenty of knowledge regarding conventional rectangular SRRs, and thus it is a well-known metasurface. In this context, the complementary rectangular SRR is quite familiar in terms of its use. Mechanical engraving was used in order to develop such MSs, using a home-built CNC router. Corresponding drawings of the MSs were developed using the open-source drawing software EASEL (Inventables Inc., Chicago, IL, USA, https://site.inventables.com/technologies/easel), which was also used to appropriately transform each drawing to corresponding g-code files that can be read by the CNC router. Then, the router moved in all three dimensions upon engraving with a thin metallic carpenter blade (diameter: Ø 0.5 mm); thus was the final engraved subject fabricated. The substrate/material used for engraving was a typical plain FR-4 surface (1 mm thick), covered with a 35 µm-thick film of pure Cu. Throughout the engraving procedure, the blade removed certain areas of the Cu film; thus the final complementary MSs were grown (see Figure 1a). In all cases, the depth of the engraved CSRRs was 0.2 mm. 

### 2.2. Optical Microscopy Experiments

All MSs were studied regarding their dimensions via optical microscopy experiments. For this reason, an optical microscope was used (AP-8 microscope, Euromex Microscopen bv., Arnhem, the Netherlands), with maximum magnification of ×80. 

### 2.3. Electromagnetic Characterization 

The EM performance of all the engraved MSs was investigated via transmission experiments in the microwave regime. In particular, all engraved CSRRs were measured using a combination of a P9372A Vector Network Analyzer (VNA) (Keysight, CA, USA) and WR187 and WR137 waveguides (Figure 2d). Details regarding the set-up and the measurement procedure were previously described [35,44,45].

### 2.4. Aqueous (Pollutants) Solution Preparation

Commercially available fertilizer (ammonium sulfate, A-SA, 21-0-0, TEOFERT, Athens, Greece) and a well-known commercially available laundry detergent (Dixan; Henkel AG & Co. KGaA, Düsseldorf, Germany) were obtained from a local agricultural shop and a supermarket, respectively, to play the pollutants’ role in the present work. Stock aqueous solutions of 0.0, 2.0, 5.0 and 10.0% *w*/*w* fertilizer and 10.0% *v*/*v* laundry detergent in tap water were prepared in order to verify the sensing capabilities of the CSRRs of this work.

### 2.5. Sensing Performance of the Fabricted MSs

The sensing properties of the fabricated MSs were investigated using the set-up described in Section 2.3. In order to explore the sensing behavior of the CSSRs, a trace quantity (~1 mm^3^) of tap water was placed into the engraved area of each CSRR using a microliter syringe suitable for dispensing volumes from 0.05 μL up to 10 μL (Hamilton Bonaduz AG, Switzerland). Then the CSRR device was put into the waveguide, and the EM response change was recorded. It should be noted that the waveguide was in a vertical position (with respect to the position pictured in Figure 2d); thus the liquid inserted into the CSRRs could not splited out of the engraved area. Similarly, the sensing properties of the CSRRs were investigated with respect to the aqueous solutions of fertilizers and detergents, as described in Section 2.4. Finally, the engraved CSRRs were also used to monitor the degradation of the above pollutants in aqueous solutions, which were subjected to photocatalysis experiments.

### 2.6. Photocatalytic Experiments 

The sensing efficiency of the engraved CSRRs was studied against the degradation of the above-mentioned stock solutions of fertilizer and laundry detergent using a heterogeneous photocatalytic approach, which employs semiconductor materials and a UV light source. This is a promising route for the removal of persistent pollutants to produce harmless end products [46]. 

First, filaments consisting of acrylonitrile butadiene styrene (ABS; INEOS Styrolution (Frankfurt, Germany)) and ZnO (Sigma-Aldrich, St. Louis, MO, USA) nanoparticles (average particle size ~50 nm) were fabricated as previously described [47]. Then, these filaments were employed in a dual-extrusion FDM-type 3D printer (Makerbot Replicator 2X; MakerBot Industries, Brooklyn, NY, USA), and polymeric grids were produced as potential photocatalytic samples. The photocatalytic activity of the 3D-printed samples was studied by means of the reduction of the stock solutions described in Section 2.4. The photocatalytic samples were placed in a custom-made quartz cell, and the entire setup (cell + solution + sample) was illuminated for up to 60 min using a UV lamp centered at 365 nm (Philips HPK 125 W) (msscientific Chromatographie-Handel GmbH, Berlin, Germany) with a light intensity of ~6.0 mW/cm^2^ [48].

### 2.7. Raman Spectroscopy Experiments 

The concentration of ammonium sulfate (degradation), which is the main constituent of the fertilizer used in this study, was monitored via room temperature Raman spectroscopy measurements. A Horiba LabRAM HR Evolution confocal micro-spectrometer (HORIBA FRANCE SAS, Longjumeau, France) was used in backscattering geometry (180°). The spectrometer was equipped with an air-cooled solid-state laser (HORIBA FRANCE SAS, Longjumeau, France) operating at 532 nm with 100 mW output power. Raman spectra were collected at 0, 20, 40, and 60 min of illumination of the corresponding fertilizer stock solution (10% *w*/*w*). Ammonium sulfate degradation was recorded based on observation of its main Raman peak in the range of 960–1000 cm^−1^. The same solutions (at 0, 20, 40, and 60 min illumination) were forwarded to the set-up described in Section 2.3 and used to explore the sensing behavior of the CSSRs following the protocol of Section 2.5.

### 2.8. UV-Vis Spectroscopy Experiments

In addition, the decolorization of laundry detergent stock solution was monitored via UV-Vis spectroscopy in absorption mode (absorption at λmax, 353 nm). In this context, a BIOBASE BK-D590 Double Beam Scanning UV/VIS Spectrophotometer (BIOBASE, Jinan, Shandong, China) was used, over the wavelength range of 190 to 1100 nm using a 1200 lines/mm grating. UV-Vis absorption data were collected at 0, 20, 40, and 60 min of stock solution illumination (10% *v*/*v*). The degradation of the detergent was recorded based on the reduction of its main absorption peak in the range of 290 to 400 nm. Once more, the detergent solutions (for 0, 20, 40, and 60 min illumination) were forwarded to the set-up described in Section 2.3 and used to explore the sensing behavior of the CSSRs following the protocol of Section 2.5

Therefore, the sensing properties of the CNC-fabricated MSs were compared using state-of-art spectroscopy techniques, such as Raman and UV-Vis spectroscopy. To ensure repeatability, the photocatalysis experiments were performed at least 3 times.

### 2.9. Theoretical Simulations

The experimental results were further supported by theoretical simulations performed with continuous wave (CW) excitation, using the frequency domain solver of the CST Studio Suite (CST Microwave Studio, Computer Simulation Technology GmbH, Darmstadt, Germany), based on the finite integration technique. The simulated MS consists of two CSRRs on a copper-coated FR-4 substrate (Figure 1b). The copper layer was modelled on an electric conductivity of 5 × 10^7^ S/m, while the relative electric permittivity of the FR-4 substrate was 4 − 0.04i (loss tangent of 0.01). Τhe waveguide walls were considered to be perfect electric conductors. The TE_10_ mode of the waveguide excited the structure in the frequency range of 3 to 9 GHz and the S-parameters were calculated using rectangular waveguide ports. In order to perform the simulations introducing water within the engraved volume of the CSRRs, the experimentally measured dispersive complex electric permittivity of water was used (i.e., see Appendix A).

## 3. Results

### 3.1. Optical Microscopy 

Figure 1a shows a typical optical microscopy picture of engraved CSRRs. In general, all MSs seemed to be rectangular and well-shaped; their lines were straight, with clean and parallel edges. Corners inside the SRRs were sharp with right angles, while the corners outside the SRRs were rounded, which is attributed to the engraving procedure as well as to the carpenter blade size. The thinner the blade size, the higher the engraving resolution obtained. Nonetheless, very thin blades are easily broken, especially when the engraved area is metallic, such as in our case. Therefore, the use of relatively thicker blades diminishes the resolution of the engraving. 

In addition, the distance between opposite SRRs was uniform: in all cases, equal to their line width. On the other hand, the MS gap was not properly formed in all cases, while its edges were quite rounded. The roundness of the gap was more pronounced when the width of the engraved SRR was narrow (i.e., see Figure 1b), while it became sharper with increases to the width of the engraved SRR (i.e., Figure 1a,b). Such a construction failure can be attributed to the thickness of the blade used. The dimensions of the fabricated CSRRs are presented in Table 1.

### 3.2. Electromagnetic Characterization 

As seen in Figure 1, a single unit cell of the engraved CSRRs was studied in this work. The EM response of the engraved CSRRs is denoted in Figure 2. In all cases, well-defined and sharp dips were observed in S_21_ vs. *f* spectra (transmission coefficient depends on the ${\left|S_{21}\right|}^{2}$). Corresponding S_11_ vs. *f* spectra (not shown here) do not exhibit any feature (reflection coefficient is proportional to the ${\left|S_{11}\right|}^{2})$. Therefore, the observed S_21_ dips are suggestive of absorption of the incident electromagnetic wave; thus, CSRRs effectively absorb EM energy at certain resonant frequencies. All minima are deep (<−25 dB) and comparable to conventionally PCB-printed SRRs [49], indicating the effective EM performance of the CSRRs. Figure 2a shows the EM response of the studied SRRs with respect to their length L. It is obvious that the resonance frequency shifts toward higher values upon decreasing L, corroborating previous experimental reports [35,49]. On the other hand, increasing the line width results in a resonance frequency shift to higher frequencies (Figure 2b), while widening of the gap leads to a resonance shift, as shown in Figure 2c. All the above-presented results are consistent with previously reported data for conventional PCB-printed SRRs. In addition, our findings were qualitatively verified via corresponding theoretical simulations (Appendix A). Therefore, it can be safely concluded that the engraved CSRRs resemble conventional PCB-printed SRRs, in terms of both resonance frequencies and transmission dips’ intensity.

### 3.3. Sensing Performance of the Fabricated MSs

Prior to the investigation of the sensing properties of the CSRRs for typical water pollutants, such as detergents and fertilizers, it is crucial to investigate whether the CSRRs possess any response to the presence of water or not. Furthermore, it is important to identify the minimum volume of water (and aqueous solutions) required to activate the CSRRs (if any). In addition, it is important to explore if there is a maximum water quantity above which the MS does not respond. In this context, we proceeded to the following experiments: 

First, the EM response of the CSRRs was measured with respect to the presence of water, which was homogeneously distributed on the engraved area of the CSRR. Typical results for the C2 sample are presented in Figure 3a (results for all the CSRRs are presented in Appendix A). It can be clearly seen that by inserting the proper amount (1 mm^3^) of water into the engraved area of the MS, a clear shift toward lower values in the resonance frequency was observed (Δ*f* ~ 97 MHz). Moreover, the corresponding dip became broader. Similar behavior could be observed in the C5 sample (L = 7.9 mm, Δ*f* ~ 270 MHz) and C11 sample (L = 6.5 mm, Δ*f* ~ 863 MHz). Thus, resonance frequency increased with decreasing MS length. On the other hand, the dip became shallower and wider. The presence of water in the CSRR engraved area, in all three cases, led to the resonant frequency decrement due to the high permittivity (real part of permittivity) of water as well as results in the dip broadening due to loss (imaginary part). The shift was stronger (in absolute and fractional terms) in smaller CSRRs, since the inserted water occupied a larger portion of the slot (see last column in Table 1) and, thus, overlapped more with the fields in the slot. Therefore, the EM response of the CSRR was affected by the presence of water. 

In addition, the performance of the CSRRs was evaluated via the determination of the quality factor $Q=f_{res}/FWHM$, the normalized sensitivity $S\;(\%)=100\times \left(f_{air}-f_{water}\right)/\left[f_{air}\left(\varepsilon_{water}-1\right)\right]$, and the figure of merit FoM=S/FWHM [50], where $f_{res}$ is the resonance wavelength in the presence of water, *Δf = f_air_* − *f_water_*, is the resonance frequency shift observed due to the presence of water in the MS, *ε*_water_ is the corresponding dielectric constant of water at resonance frequency, and *FWHM* is the full width at half maximum of the shifted dip. All *Q*, *S* and *FoM* values are presented in Table 2. In addition, dielectric permittivity measurements for water are also presented (Appendix A), from which the refractive index could be calculated with respect to the frequency. It is obvious that larger CSRRs exhibit larger *Q* and *FoM* values, suggesting that they are more efficient sensors. However, this is only partially reasonable. As we show later, the water volume inserted into the engraved area is a crucial parameter to achieve reliable sensing capability. Regarding *Q* factor values, they are comparable to other CSRRs dedicated to liquid sensing [51,52,53]. Moreover, although low, S values are also comparable to others reported in the literature for microwave sensors [52,53,54]. In addition, *FoM* values are very small, which is mainly attributed to the low sensitivity S values obtained for water. 

After confirming the EM response of the CSRRs in the presence of water, we proceeded to another experiment in which the EM behavior of the studied MSs was investigated with respect to the volume of water inserted into the engraved area. In particular, a measured volume of water was uniformly distributed into the engraved CSRRs, and the EM response was measured. As the volume of water inserted into the CSRRs was varied, corresponding EM spectra were recorded. Figure 4a shows typical EM spectra of the C6 sample. By increasing the volume of the water introduced into the engraved area, the resonance’s frequency shifted toward lower values. Moreover, the minimum S_21_ value decreased monotonically with increasing water volume, and for water volumes above ~5 mm^3^, the resonance almost vanished, most likely due to the high imaginary part of water. Interestingly, the volume of the engraved area of the C6 sample was calculated to be ~6.60 mm^3^. Corresponding theoretical simulations qualitatively verified the above-described results (i.e., Appendix A), indicating high sensitivity of the MS to the presence of water. Therefore, it can be proposed that resonance is quenched when the engraved area is full of water. Moreover, it can obviously be anticipated that the volume of water inserted into the CSRR is crucial for its sensing performance. In particular, resonance frequency shift increases with increasing water volume (Figure 4b). On the other hand, S_21_ intensity decreased (Figure 4b), while the dips broadened. Corresponding analysis (Appendix A) clearly shows that the CSRR performed at its maximum *FoM* for water volumes up to ~2 mm^3^. Above that volume, *FoM* decreased, suggesting that the C6 sample exhibited its most efficient performance when the water volume inserted into the engraved area was less than 2 mm^3^. Similar calibration was performed for all studied CSRRs. It was found that the threshold water volume for samples C1, C2, and C3 (L = 10 mm) was ~3 mm^3^, while for the sample C11 (L = 6 mm) it was ~1 mm^3^. For samples C4 and C5, the upper volume limit was ~2 mm^3^. 

The above-described experiments reveal the sensing performance of the fabricated CSRRs in the presence of water. Due to its high dielectric constant, water severely affected the EM response of the CSRRs, resulting in the considerable resonance shift observed, regardless of MS size. Thus, it becomes prudent to explore the performance of the CSRRs, in the presence of polluted water, i.e., water contaminated with ammonium sulfate fertilizers. In this context, water solutions containing various concentrations of fertilizer (2%, 5%, and 10% wt.) were prepared and injected into the engraved area of the CSRRs; corresponding EM spectra are presented for the C4 sample (Figure 5a).

It can be clearly observed that resonance frequency shifted toward higher values as the concentration of fertilizer increased (Figure 5b). Moreover, corresponding S_21_ intensity significantly decreased with increasing fertilizer concentration (Figure 5c). Correspondingly, the solution’s dielectric constant decreased with increasing concentration (i.e., Appendix A). Thus, the shift toward higher frequencies mainly resulted from the dielectric constant decrease. Therefore, the presence of ammonium sulfate fertilizer, even in traces, directly affected both the resonance frequency and the intensity of the minimum of the MS, enabling its sensing capability against water polluted with fertilizers. The corresponding quality factor, sensitivity, and *FoM* values are 10.5, 10.8%, and 18.5 respectively. Here it must be noted that these values were calculated with respect to pure water; therefore, the corresponding relations mentioned above have been appropriately modified, i.e., $S=100\times \left(f_{cont}-f_{water}\right)/\left[f_{water}\left(\varepsilon_{water}-\varepsilon_{cont}\right)\right]$, where *f_cont_*, *ε_cont_* is the resonance frequency and the dielectric constant of the contaminated water, respectively. 

The above-described behavior was further investigated by exploring the EM response of the engraved CSRRs with respect to the degradation of pollutants via a photocatalytic procedure. In this context, an aqueous solution containing fertilizer (10% wt) was used to resemble polluted water. Photocatalytic experiments were performed using 3D-printed photocatalytic structures, containing ZnO, as previously described [41]. The degradation of the pollutant was monitored via Raman spectroscopy. The photocatalytic procedure lasted for 1 h, and corresponding spectra were obtained every 20 min. Each time, a tiny amount (~1 mm^3^) of the solution was transferred to the CSRR, and the EM response of the later was measured accordingly. Corresponding results for the aqueous solution including the fertilizer (C4 sample) are presented in Figure 6. 

It can clearly be seen (Figure 6a) that the resonance frequency shifted to lower values as the time of photocatalysis increased, approaching the resonance frequency of pure water (blue panel of Figure 6a). Moreover, the intensity of the resonance was suppressed toward the intensity of pure water (yellow panel in Figure 6a). The photocatalytic procedure resulted in the degradation of the pollutant into the solution and therefore the concentration of the pollutant (e.g., the fertilizer in our case) into the aqueous solution decreased. Thus, the solution became more and more diluted, which was demonstrated by both the resonance frequency shift toward the resonance frequency of pure water, as well as the S_21_ intensity approaching that of pure water as the photocatalytic procedure occurred. Interestingly, after 60 min, the resonance frequency of the solution almost equaled that of pure water, probably indicating the complete degradation of the fertilizer. Accordingly, Raman spectra (Figure 6b) show the decrement of the characteristic peak of the ammonium sulfate (detailed analysis of the Raman spectra of the ammonium sulfate is presented in Appendix A), which is the main chemical ingredient of ammonium sulfate fertilizers. Such a decrease is in agreement with the results presented in Figure 6a. Therefore, the studied CSRRs can be effectively used as microwave sensitive indicators in photocatalytic systems and devices for water pollution from fertilizers. Correspondingly, quality factor *Q*, normalized sensitivity *S,* and *FoM* were found to be 9.75, 11.9% and 22.4, respectively. 

Subsequently, the EM response of the engraved CSRRs with respect to the degradation of an aqueous solution containing laundry detergent (10% *v*/*v*) was investigated. In this case, the pollutant’s concentration was recorded by means of UV-VIS spectroscopy as previously reported [41]. Experimental results are shown in Figure 7. In particular, a clear shift toward lower frequencies was observed (Figure 7a) as the time of the photocatalytic procedure increased. Interestingly, after 60 min, the resonance frequency of the solution was equal to that of pure water, indicating the degradation of the pollutant via the photocatalytic process. Additionally, the corresponding UV-Vis spectra (Figure 7b) exhibited a distinct reduction in the characteristic peak intensity with respect to time, suggesting the degradation of the pollutant into the solution. Therefore, the UV-Vis experimental data came to significant agreement with EM experimental results. This is a strong indication that engraved CSRRs can be utilized as sensors in photocatalytic devices and systems for the degradation of detergents in the water. Notably, the calculated *Q*, *S* (*%*), and *FoM* values are 11.2, 16.2% and 34.3, respectively. 

## 4. Summary and Conclusions

In this study, mm-scale rectangular CSSRs were fabricated, utilizing the CNC engraving method. Several CSRRs were developed with various dimensions regarding their length, line width, and gap. Macroscopic observation revealed that all the fabricated CSRRs exhibited well-shaped and uniform metasurfaces with uniform line thickness; however, outside corners were rounded. The measured dimensions of the metasurfaces were quite close to the desired ones. Therefore, high-quality (in terms of design and dimensions) CSRRs can be developed by employing the CNC engraving method. 

The fabricated metasurfaces were extensively investigated regarding their electromagnetic behavior. It was found that all CSRRs exhibited a distinct S_21_ dip at a certain frequency, indicative of absorption of the incident microwave signal. The observed resonance frequency is directly affected by the dimensions of the CSSR, i.e., its length, line width, and gap. Despite its inverse structure, CSSR behavior is similar to that of conventional SRRs of the same dimensions.

The introduction of water into the engraved area of the CSRRs directly affected the EM response of the metasurfaces. In particular, resonance frequency shifted toward lower values while the S_21_ increased as the volume of the water introduced into the CSRR increased. Moreover, when the water volume reached the volume of the engraved area, S_21_ vanishes. 

Additionally, the developed CSRRs were tested against aqueous solutions containing traces of pollutants such as fertilizers and detergents. All of them exhibited a remarkable resonance shift along with S_21_ decrement in the presence of pollutants. Furthermore, both resonance shift and S_21_ increment changed monotonically the concentration of the pollutant in the aqueous solution changed. Thus, the developed CSRRs show a noteworthy sensitivity with respect to fertilizers and detergents and could potentially be used as sensors of water pollution from agriculture and sewage.

Finally, the EM response of the produced CSRRs was compared to products from the photocatalytic procedure. In this way, the sensing properties of the fabricated CSRRs were compared with state-of-art spectroscopy techniques, such as Raman and UV-Vis spectroscopy. In particular, aqueous solutions containing traces of fertilizers and detergents were photo-catalyzed so the pollutant material decomposed, resulting in the decrement of the concentration of the pollutant in the water. The CSRRs exhibited a clear resonance shift toward lower frequencies as the degradation of the pollutants evolved, approaching the resonance frequency of pure water. Therefore, CSRRs could be effectively used as microwave sensors in photocatalytic systems and devices.

In conclusion, engraved CSRRs could be used as high-resolution microwave sensors for observing/controlling/manipulating water quality in terms of two of the most commonly used pollutants, namely fertilizers and detergents.

## Figures and Tables

**Figure 1 materials-16-05290-f001:**
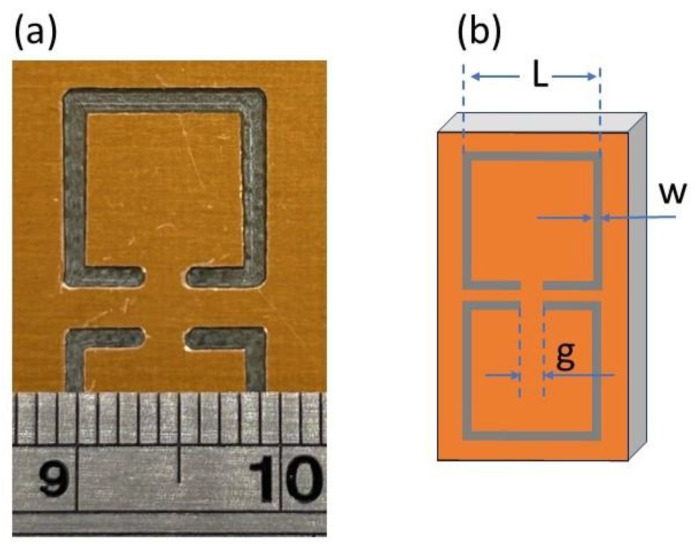
Optical microscopy photographs for (**a**) sample with dimensions as follows: length, L = 9.9 mm; width, w = 1.1 mm; and gap; g = 1.2 mm and (**b**) sketch of the CSRR structure and corresponding dimensions. The distance between the two SRRs is equal to the line width.

**Figure 2 materials-16-05290-f002:**
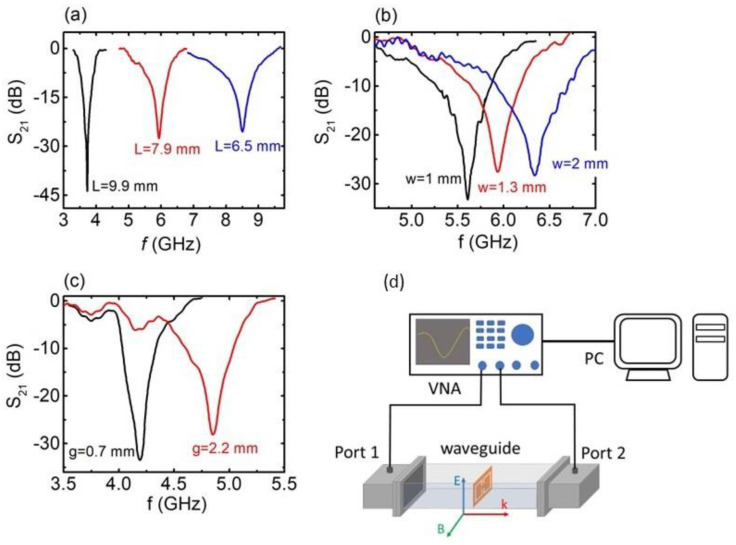
S_21_ vs. *f*, for the engraved CSRRs with respect to (**a**) the metasurface (MS) length L (black, red, and blue lines correspond to C2, C5, and C11 samples, respectively), (**b**) the width of the MS (black, red, and blue lines correspond to C4, C6 and C10 samples, respectively) and (**c**) the dimension of the gap g (black and red lines correspond to C1 and C3 samples, respectively). (**d**) Experimental set-up drawing for the measurement of the electromagnetic (EM) response of the engraved MSs. The orientation of the MS into the waveguide is also shown.

**Figure 3 materials-16-05290-f003:**
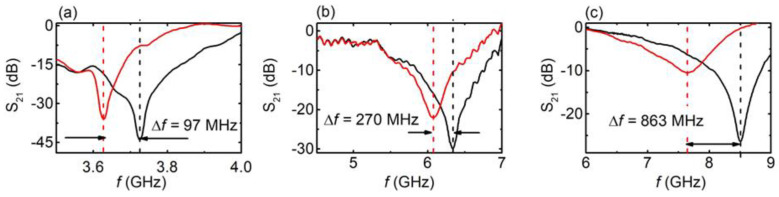
S_21_ vs. *f*, for engraved CSRRs with (solid red line) and without (solid black line) the presence of water in the engraved area. (**a**) C2 sample, (**b**) C5 sample, and (**c**) C11 sample.

**Figure 4 materials-16-05290-f004:**
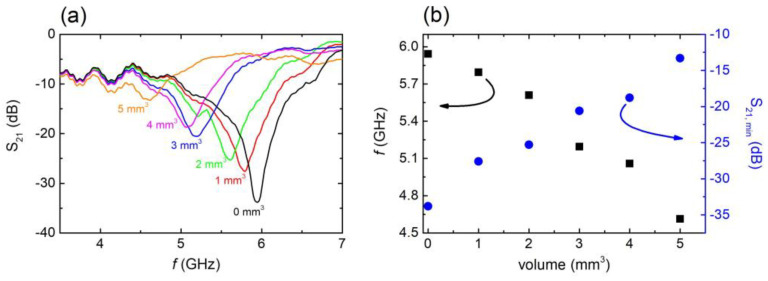
(**a**) S_21_ vs. *f*, for C6 sample with several water volumes introduced into the engraved area. (**b**) Resonance frequency (black squares) and minimum S_21_ values (blue circles)with respect to the volume of the water introduced into the engraved area.

**Figure 5 materials-16-05290-f005:**
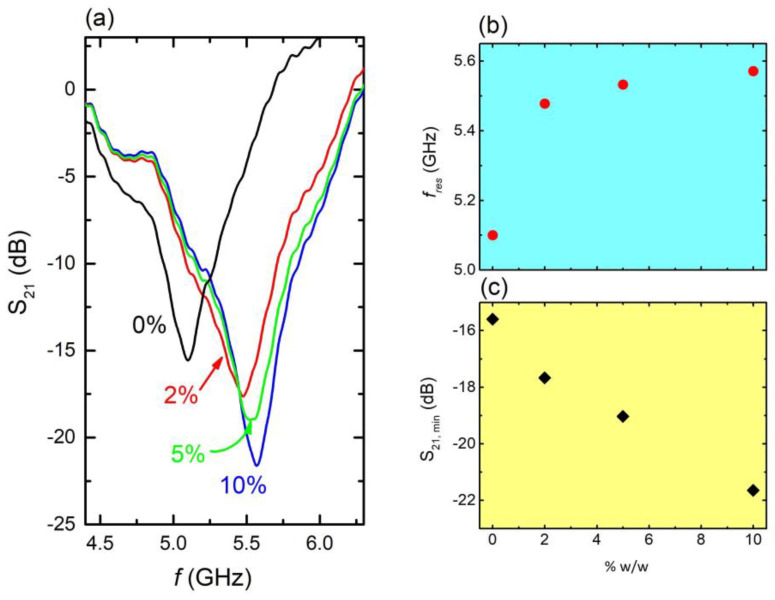
(**a**) S_21_ vs. *f*, on several aqueous solutions including fertilizers as contaminants at various concentrations, i.e., 10% *w*/*w* (blue line), 5% w/w. (green line), 2% *w*/*w* (red line), and 0% (black line—pure water) (C4 sample). (**b**) Resonance frequency *f_res_* vs. solution concentration as extracted from panel (**a**,**c**). Corresponding S_21_ intensity as a function of solution concentration.

**Figure 6 materials-16-05290-f006:**
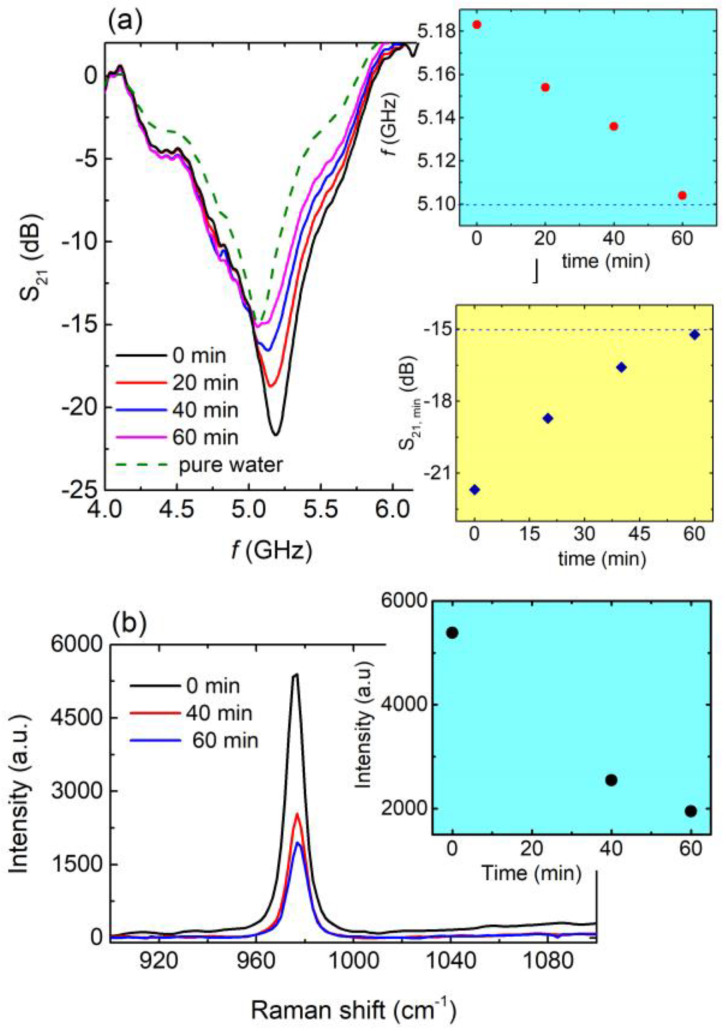
(**a**) Main panel: S_21_ vs. *f*, for 0 min (black line), 20 min (red line), 40 min (blue line), and 60 min (magenda line) of photocatalysis for the aqueous solution of fertilizer. Corresponding S_21_ (*f*) curve for pure water (green dash line) also included for comparison. Blue panel: Resonance frequency with respect to photocatalysis time as extracted from main panel. Blue dashed line refers to the resonance frequency of pure water. Yellow panel: S_21,min_ values as a function of time as extracted from main panel. Blue dashed line corresponds to the S_21,min_ value of pure water. (**b**) Main panel: Raman spectra for 0 min (black line), 40 min (red line), and 60 min (blue line) of photocatalysis regarding the aqueous solution of fertilizer. Yellow panel: Peak intensity with respect to photocatalysis time as extracted from main panel.

**Figure 7 materials-16-05290-f007:**
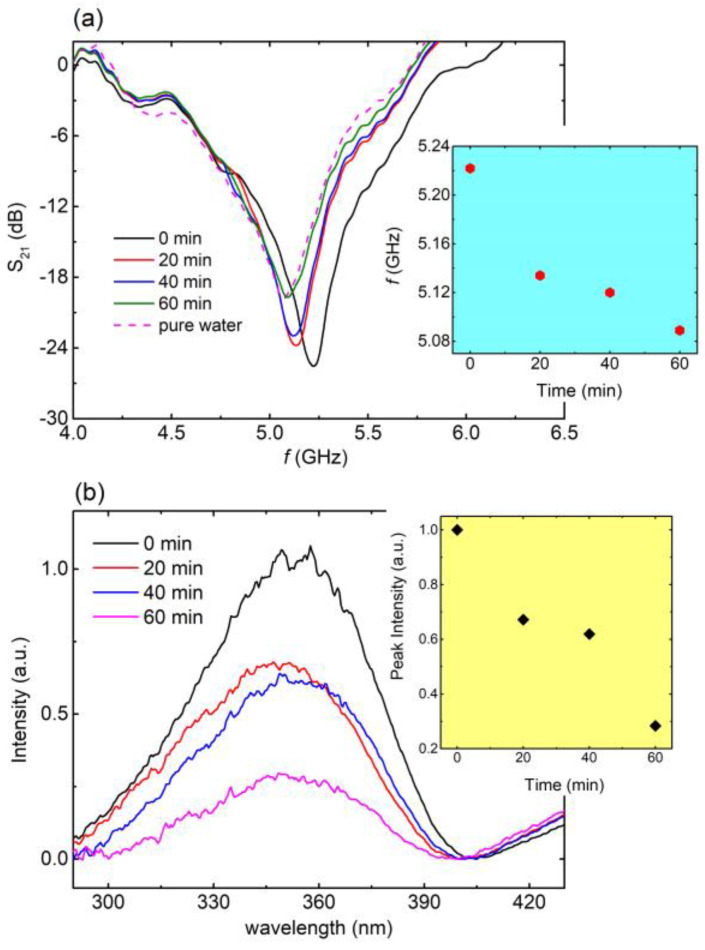
(**a**) S_21_ vs. *f*, for 0 min (black line), 20 min (red line), 40 min (blue line), and 60 min (green line) of photocatalysis for the aqueous solution including laundry detergent. Corresponding S_21_ (*f*) curve for pure water (magenta dashed line) is also included for comparison. Blue panel: Resonance frequency with respect to photocatalysis time as extracted from main panel. (**b**) UV-VIS spectra for 0 min (black line), 20 min (red line), 40 min (blue line), and 60 min (magenta line) of photocatalysis, regarding the aqueous solution of laundry detergent. Yellow panel: Peak intensity, with respect to the photocatalysis time, as extracted from main panel.

**Table 1 materials-16-05290-t001:** Measured dimensions and calculated engraved volume for all studied CSRRs (engraving depth is 0.2 mm in all cases).

CSRR Name	L (mm)	w (mm)	g (mm)	V (mm^3^)
C1	9.9 ± 0.1	1.2 ± 0.1	0.7 ± 0.1	10.9
C2	9.9 ± 0.1	1.1± 0.1	1.2 ± 0.1	7.39
C3	9.9 ± 0.1	1.1± 0.1	2.2 ± 0.2	7.26
C4	7.9 ± 0.1	1.0 ± 0.1	1.2 ± 0.2	5.28
C5	7.9 ± 0.1	1.1 ± 0.1	1.1 ± 0.1	5.24
C6	7.9 ± 0.1	1.3 ± 0.1	1.0 ± 0.1	6.60
C10	8.0 ± 0.1	2.0 ± 0.1	1.2 ± 0.2	5.92
C11	6.5 ± 0.1	1.1 ± 0.1	1.3 ± 0.2	4.66

**Table 2 materials-16-05290-t002:** Resonance frequency *f*_res_, frequency shift Δ*f*, dielectric constant *ε*_water_ of water at resonance frequency, full width at half maximum (*FWHM*), quality factor *Q*, normalized sensitivity *S*, and figure of merit (*FoM*) for all engraved CSRRs.

CSRR Name	*f_res_* (GHz)	Δ*f* (MHz)	*ε_water_*	*FWHM* (MHz)	*Q*	*S* (%)	*FoM* (10^−4^)
C1	4.053	134	76.3	177	30.2	0.042	2.40
C2	3.627	97	77.0	40	90.7	0.034	8.57
C3	4.200	200	76.1	216	19.4	0.060	2.80
C4	5.080	400	74.7	147	34.5	0.099	6.74
C5	6.070	270	72.8	147	14.6	0.059	4.03
C6	5.490	526	74.0	487	11.3	0.120	2.46
C10	5.627	114	73.7	343	16.4	0.028	0.796
C11	7.643	863	69.2	1019	7.50	0.149	1.46

## Data Availability

No data are available.

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
