# Peer review of "Fabrication of mm-Scale Complementary Split Ring Resonators, for Potential Application as Water Pollution Sensors"

_materials, 2023, doi:10.3390/ma16155290_

Round 1
Reviewer 1 Report
Report on
Fabrication of mm-scale complementary split ring resonators, for potential application as water pollution sensors
written by
Zacharias Viskadourakis, George Fanourakis, Evangelos Tamiolakis, Anna Theodosi, Klytaimnistra Katsara, Nikolaos Rafael Vrithias, Odysseas Tsilipakos and George Kenanakis
The paper is written about complementary split ring resonators that are proposed for water pollution sensors. Split ring resonators are well known elements of metamaterials, and their widely tuneable properties makes them suitable for sensor development. Mitigation of water pollution is a great demand, therefore development of metamaterial sensors capable of indicating presence of harmful chemicals with a concentration sensitivity is an important forward step towards application of metamaterials. The paper is valuable for publication after revision realized considering the following questions and suggestions.
1. The introduction would be more convincing, if the first six references to websites would be replaced by journal papers like [7].
2. The illumination methodology could be extrapolated based on Fig. 2c, but there is no information about the polarization direction and the identification of S21 (transmission) and S11 (reflection) ports that would be necessary to reproduce the results (as in the references # 35 & 41).
3. The CNC method appears already in the abstract, but this abbreviation as well as the EASEL has to be defined, and a reference to the methods should be inserted.
4. In the introduction section the main results are usually summarized, but it is unusual to detail the methods, since it suggests that the novelty is in the methodological approach.
5. In section 2.7 the description of the UV-visible and Raman spectroscopy is mixed.
6. The correct term to qualify the sensors is the FOM, defined as the ratio of the frequency (wavelength) shift and refractive index modification (RIU=dn/dn), not the “relative tunability”.
7. Complementary resonators result in complementary responses, e.g. one would expect that dips in transmittance for SRR are complementary with dips in reflectance in CSRR. What is the meaning of the S21 in the paper?
8. In the definition of the relevant FOM it is reasonable to consider the FWHM of the resonance curves and its modification during filling with water.
9. The resonance curves broaden and disappear, when the amount of material is increased. The authors should address the existence of an upper bound and an optimal amount for detectable material in case of different compositions.
10. The amount and the concentration dependent dielectric parameter have reversal effect, that indicates again the existence of trade-off, and the necessity of a third parameter (e.g.) FWHM monitoring to distinguish the cases, when a certain amount of pure water and a different amount of water with added detergent result in a dip at the same frequency.

Reviewer 2 Report
In this work, the authors present an interesting approach to measuring water pollutants using a complementary split ring resonator CSRR metasurface created using the CNC engraving method. There were a total of 9 CSRRs created, and frequency shifts and intensities were measured and compared when introduced to polluted water containing fertilizers and detergents of various percentages. These measurements were verified with Raman spectra of the same samples. The presentation for this work is comprehensive, well done, and includes convincing results.
A few remarks:
Line 158, there is a typo; I believe it should CSRRs, not CSRRS.
Line 162 Section 2.6 Photocatalytic experiments. I found this section a little confusing, especially with the fabrication of the ZnO enriched polymeric grids. Were two filaments used to create this grid, or was the ZnO dopped into the filament before printing? There is mention of the need for a dual extruder, and I was hoping the authors could expand more on this setup.
Figure 1. I was curious why the aspect ratio was changed between the two images. Was this intentional or a consequence of the microscope being used?
Line 243. The authors list the different CSRR configurations. I was curious about what the motivation was behind choosing the ones presented.
To summarize, the paper is quite interesting and well-formulated and has my recommendation for publication. But the authors should take into account the comments above.
Reviewer 3 Report
The author proposed a sensor based on mm-scale complementary split ring resonators (CSRR) metasurface that can distinguish the several ammonium sulfate fertilizer as water contaminant. The tunability of resonance frequencies is used to determine the concentration of water contaminant. The design and experiment are presented clearly in this manuscript. However, several similar works related to sensors based on CSRR metasurface have been published [Micromachines 14, no. 2 (2023): 384, IEEE Sensors Journal 18, no. 24 (2018): 9978-9983, and IEEE Sensors Journal 19, no. 24 (2019): 11880-11890]. The theoretical and numerical in characterizing the sensor are not present in this work. Therefore, I recommend that the manuscript be accepted after major revision. My comments are listed as follows:
1. What is the novelty of this work apart from previous work? Could the author put the author’s aims in the introduction section?
2. This work demonstrates the point of view of experimental-based. However, the numerical of the nearfield and the current distribution at its resonance frequency is interesting to study.
3. The design of CSRR is not clearly presented in Figure 1(c). The manuscript does not present the substrate’s thickness, the unit cell period, and the distance of SRRs facing each other. The author should present the 3D model of CSRR, including the substrate and engraved CSRR.
4. Figure 1(b) seems to tilt with an angle. Does the sample also have a tilt when it is measured? The author should display a clear image.
5. The sensing parameter such as sensitivity, Q-factor, and figure of merit (FOM) should be demonstrated in Figures 5 and 6.
6. The setup experiment that involved water pollution should display in the manuscript.
7. To help the readers have a more comprehensive understanding of the new research on metasurfaces, I suggest supplementing some latest works about biosensors with large refractive-index sensitivities [Photonics Research Vol. 10, Issue 9, pp. 2215-2222 (2022)], dual-band multifunctional coding metasurfaces [Photonics Research 10, no. 2 (2022): 416-425] and terahertz intensity modulators [Optics Express 28, no. 19 (2020): 27676-27687].
Reviewer 4 Report
The papers is dedicated to design, fabrication and characterization of split-ring resonators as potential sensors of water pollution. It is well written, well structured and can be interesting for reader. Its publication should be ensured but some revisions are needed prior to publication.
The following issues should be addressed in the revised manuscript. 1) Please explain why the design comprising two CSRRs has been selected. Why not three or one (C)SRR; why two CSRRs are crucial? It is recommended to discuss the design from the point of view of subwavelength resonators. In particular, earlier works in which more than one (C)SRR per unit cell is used should be cited and briefly discussed, e.g., see J. Appl. Phys. 113, 084903 (2013); IEEE Trans. Antennas Propag. 68, 5071 (2020); J. Appl. Phys. 101, 014909 (2007). Besides a siminal paper on CSSRs should be cited and briefly discussed: Phys. Rev. Lett. 93, 197401 (2004). 2) The role of absorption should be discussed in more detail. It looks that the increase of the water volume, like in Fig. 4, results in increase of S21 rather because of higher overall losses. This can also be expected to occur based on the permittivity behavior shown in Fig. S1. Indeed, the imaginary part of permittivity is large regardless of whether 0% or 10% is taken. Therefore, possible limits of the method that are resulted from the effect of losses should be discussed. 3) More details about the results in Fig. 3 can be given. 4) A comment on applicability of the proposed sensor for different kinds of pollution is needed. 5) The legends in the figures should be double checked. For instance, the meaning of 0%, 1%, 2%, 5% and 10% in Fig. S1(b) should be clearly indicated.
Round 2
Reviewer 3 Report
The author has made a great effort to reply to the reviewer’s comments. The novelty of this work, apart from previous works, is also described in the manuscript. However, some of the comments are still not answered clearly. Therefore, I recommend that the manuscript be accepted after major revision. My comments are listed as follows:
1. The author mentions that the simulation of the near-field distribution and current distributions of proposed metasurfaces are present in supplementary Figures S2 and S8. However, I did not see any near-field or current distributions of the proposed metasurfaces in those Figures.
2. Based on comment 3, the author mentions the proposed metasurface has dimensions L = 9.9 mm, w = 1.1 mm and g = 1.2 mm in Figure 1. However, the period of the unit cell and the gap distance between SRRs facing each other are not mentioned in the manuscript content. The detailed geometrical parameters should be better mentioned in this manuscript.
3. The comparison of Q factor, sensitivity and FOM to the previous works should be better presented and put into a table.
4. Based on comment 6, the author’s reply should be put in the sub-section of 3.3. However, the author has a statement that the setup is similar to Figure 2(d). Then, how did the author inject the liquid sample of water pollution into the waveguide? Could the author describe the measurement steps in detail?
5. Figures 5(a) and 5(b) present a significant frequency shift when the concentration of contaminants is increased from 0% to 2%. Could the author discuss the reason for the significant frequency shit? Based on that significant shift, could the author detect the contaminants with a concentration of 1%?
6. I did not see any revision of reference in this manuscript. To help the readers have a more comprehensive understanding of the new research on metasurfaces, I suggest supplementing some latest works about metalens [Photonics Research Vol. 10, Issue 4, pp. 886-895 (2022)], continuously tunable intensity modulators [Optics Express 28, no. 19 (2020): 27676-27687], and biosensors with large refractive-index sensitivities [Photonics Research Vol. 10, Issue 9, pp. 2215-2222 (2022)]

Author Response
please read the attached file
